# Adverse childhood experiences and resilience among adult women: A population-based study

**Hilda Björk Daníelsdóttir[1,2]\***, Thor Aspelund[1], Edda Bjork Thordardottir[1], Katja Fall[3,4], Fang Fang[4], Gunnar Tómasson[1], Harpa Rúnarsdóttir[1], Qian Yang[2], Karmel W Choi[5,6], Beatrice Kennedy[7], Thorhildur Halldorsdottir[1,8], Donghao Lu[4,6], Huan Song[9], Jóhanna Jakobsdóttir[1], Arna Hauksdóttir[1], Unnur Anna Valdimarsdóttir[1,2,6]

[1]Centre of Public Health Sciences, Faculty of Medicine, University of Iceland, Reykjavík, Iceland; [2]Department of Medical Epidemiology & Biostatistics, Karolinska Institutet, Stockholm, Sweden; [3]Clinical Epidemiology and Biostatistics, School of Medical Sciences, Örebro University, Örebro, Sweden; [4]Institute of Environmental Medicine, Karolinska Institutet, Stockholm, Sweden; [5]Department of Psychiatry, Massachusetts General Hospital, Boston, United States; [6]Harvard T.H. Chan School of Public Health, Boston, United States; [7]Department of Medical Sciences, Molecular Epidemiology and Science for Life Laboratory, Uppsala University, Uppsala, Sweden; [8]Department of Psychology, Reykjavík University, Reykjavik, Iceland; [9]West China Biomedical Big Data Center, West China Hospital, Sichuan University, Chengdu, China

**\*For correspondence:**
hilda.bjork.danielsdottir@ki.se

**Competing interest:** The authors declare that no competing interests exist.

## Abstract

**Background:** Adverse childhood experiences (ACEs) have consistently been associated with elevated risk of multiple adverse health outcomes, yet their contribution to coping ability and psychiatric resilience in adulthood is unclear.

**Methods:** Cross-sectional data were derived from the ongoing Stress-And-Gene-Analysis cohort, representing 30% of the Icelandic nationwide female population, 18–69 years. Participants in the current study were 26,198 women with data on 13 ACEs measured with the ACE-International Questionnaire. Self-reported coping ability was measured with the Connor-Davidson Resilience Scale and psychiatric resilience was operationalized as absence of psychiatric morbidity. Generalized linear regression assuming normal or Poisson distribution were used to assess the associations of ACEs with coping ability and psychiatric resilience controlling for multiple confounders.

**Results:** Number of ACEs was inversely associated with adult resilience in a dose-dependent manner; every 1SD unit increase in ACE scores was associated with both lower levels of coping ability ($\beta = -0.14$; 95% CI-0.15,–0.13) and lower psychiatric resilience ($\beta = -0.28$; 95% CI-0.29,–0.27) in adulthood. Compared to women with 0 ACEs, women with ≥5 ACEs had 36% lower prevalence of high coping ability (PR = 0.64, 95% CI 0.59,0.70) and 58% lower prevalence of high psychiatric resilience (PR = 0.42; 95% CI 0.39,0.45). Specific ACEs including emotional neglect, bullying, sexual abuse and mental illness of household member were consistently associated with reduced adult resilience. We observed only slightly attenuated associations after controlling for adult socioeconomic factors and social support in adulthood.

**Conclusions:** Cumulative ACE exposure is associated with lower adult resilience among women, independent of adult socioeconomic factors and social support, indicating that adult resilience may be largely determined in childhood.

**Funding:** This work was supported by the European Research Council (Consolidator grant; UAV, grant number 726413), and the Icelandic Center for Research (Grant of excellence; UAV, grant number 163362-051). HBD was supported by a doctoral grant from the University of Iceland Research Fund.

## Editor's evaluation

This study aims to investigate the impact of adverse experiences during childhood on adult psychological and psychiatric resilience. Leveraging the excellent data from an ongoing cohort study on Icelandic women, the authors showed that in the face of accumulated adverse childhood events the prevalence of resilience declines, which supports earlier studies suggesting that resilience is not invincibility. The study makes an important contribution to raising awareness of the adverse childhood experiences and their impact on resiliency.

## Introduction

Exposure to adverse childhood experiences (ACEs), including child abuse, neglect and growing up in dysfunctional households, is associated with elevated risk of a wide range of physical and mental health problems across the life course (*Anda et al., 2006*; *Bellis et al., 2015*; *Hughes et al., 2017*; *Petruccelli et al., 2019*). The results from a recent meta-analysis (*Hughes et al., 2017*) suggest that the adult health outcomes most clearly associated with ACEs include problematic alcohol consumption and substance use, violence, and mental illness. However, there is great variation in long-term outcomes of children exposed to ACEs, and many children remain healthy despite excessive ACE exposure. Importantly, it has been documented that a substantial proportion of individuals exposed to ACEs do not develop mental illness in adulthood (*Green et al., 2010*; *Kessler et al., 2010*), but instead exhibit resilience (*Holmes et al., 2015*; *DuMont et al., 2007*).

Resilience is generally conceptualized as maintained mental health or positive adaptation despite trauma exposure (*Luthar et al., 2000*; *Kalisch et al., 2017*; *Choi et al., 2019*; *Rutter, 2006*). And although the scientific investigation of resilience can be traced back to the 1970s (*Luthar et al., 2015*; *Masten et al., 2008*; *Garmezy, 1974*; *Garmezy, 1990*; *Werner, 1989*; *Rutter, 1990*), the complexity of the concept has to date contributed to varying definitions and measurement approaches (*Southwick et al., 2014*). Two common contemporary approaches to operationalize the concept among adults, define resilience as *perceived coping ability* reflecting individuals' perceptions of their ability to cope effectively with stress and adversity (*Campbell-Sills and Stein, 2007*; *Connor and Davidson, 2003*), and as *psychiatric resilience* reflecting an empirically derived outcome, such as the absence of PTSD or other psychiatric disorders among individuals exposed to traumatic events (*Nishimi et al., 2021*; *Sheerin et al., 2018*). Indeed, it is important to note that psychiatric resilience is never directly measured as it consists of two separate components, trauma exposure and positive adaptation, and is therefore indirectly ascertained based on evidence of the two components (*Luthar, 2006*). The different resilience definitions (i.e. perceived coping ability and psychiatric resilience) are not mutually exclusive, but rather complementary and may capture different underlying dimensions of resilience (*Choi et al., 2019*; *Sheerin et al., 2018*; *Fisher and Law, 2020*).

Only a handful of previous studies have addressed the association between ACEs and adult resilience. Childhood maltreatment, variously defined, has been negatively associated with psychiatric resilience (*Mersky and Topitzes, 2010*; *Topitzes et al., 2013*; *McGloin and Widom, 2001*; *Williams et al., 2006*) and perceived coping ability (*Campbell-Sills et al., 2009*; *Nishimi et al., 2020*) in adulthood. However, most studies have focused solely on childhood maltreatment or included only a small number of ACEs. Therefore, to date, little is known about the association between cumulative ACE exposure and adult resilience, and whether specific ACE types are to a varying extent associated with resilience.

Leveraging a large nationwide study of Icelandic women, we aimed to investigate the association between the cumulative number of a broad spectrum of ACEs and two distinct measures of adult resilience, that is perceived coping ability and an outcome-based measure of low psychiatric morbidity.

## Materials and methods

### Study sample

In this study, we utilized data from the ongoing Stress-And-Gene-Analysis (SAGA) cohort, a population-based study in Iceland on the impact of trauma on women's health. All 18–69 year-old Icelandic speaking women residing in Iceland with an identifiable address or telephone number (n≈104,197), were invited to participate in the study from March 2018. Data collection continued until July 1st 2019, yielding a total of 30,403 participating women (30% of eligible women). The participants of the SAGA cohort represent the general Icelandic female population in terms of distribution of age, education level, geographical location, and monthly wages (*Appendix 1—figure 1*).

Since trauma exposure is intrinsic to psychiatric resilience, the analytic sample was restricted to women reporting a worst traumatic event at some point during their lifetime (see description of PCL-5 below). In addition, women who had more than 25% missing on perceived coping ability (n = 511), psychiatric resilience (n = 1040) and/or ACE-IQ (n = 93) were excluded, which resulted in a final study population of 26,198 women (*Appendix 1—figure 2*).

### Measures

#### Adverse childhood experiences (ACEs)

ACEs were measured with a modified version of the Adverse Childhood Experiences International Questionnaire (ACE-IQ) developed by the WHO (*WHO, 2021*). The instrument consists of 39 items assessing how often individuals were exposed to the following 13 ACEs during the first 18 years of their life: emotional neglect, physical neglect, emotional abuse, physical abuse, sexual abuse, domestic violence, living with a household member who abuses drugs and/or alcohol, living with a household member who is mentally ill or suicidal, incarceration of a household member, parental death or separation/divorce, being bullied, witnessing community violence, and exposure to war/collective violence. Response options varied between items and items were either answered on a 5-point scale ranging from 0 (never) to 4 (always), on a 4-point scale ranging from 0 (never) to 3 (many times) or answered dichotomously 0 (no) and 1 (yes). For an overview of included items and their response options see *Appendix 1—table 1*. The recommended frequency scoring system (*WHO, 2021*), which takes into account the level of exposure for each ACE, was used to generate three types of exposure variables: (1) a continuous ACE-IQ total score ranging from 0 to 13, reflecting the number of ACEs participants were exposed to; (2) the total score was categorized (0, 1, 2, 3–4 and ≥5 ACEs) based on the distribution of the sample; (3) binary variables for each individual ACE type (described above), coded as 0 (unexposed) and 1 (exposed).

#### Perceived coping ability

Perceived coping ability was assessed with the 10-item version of the Connor-Davidson Resilience Scale (CD-RISC-10) (*Campbell-Sills and Stein, 2007*). The scale, which measures individuals' perceptions of their ability to cope effectively with stress and adversity, such as the ability to adapt to change, achieving goals despite obstacles, and maintaining positivity in the face of stress, has demonstrated good reliability and validity (*Campbell-Sills and Stein, 2007*). All items were answered on a 5-point scale ranging from 0 (not true at all) to 4 (true nearly all the time). Items were summed to create a total score ranging from 0 to 40 with a higher score indicating higher levels of perceived coping ability. As there is no standardized cut-off for the CD-RISC, the total scores were divided into quintiles (for more details see the Appendix 1), and a binary variable was created where the highest quintile was used to define a high level of perceived coping ability (i.e. resilience, CD-RISC score ≥35).

#### Psychiatric resilience

Consistent with previous literature (*Choi et al., 2019*; *Nishimi et al., 2021*; *Stein et al., 2019*) psychiatric resilience was defined as absence of or low psychiatric morbidity among women exposed to lifetime trauma, that is total sum of the inverse number of above-threshold symptom levels on PTSD, trauma-related sleep disturbances, binge drinking, depression, and anxiety.

The PTSD Checklist for DSM-5 (PCL-5) is a 20-item valid instrument that assesses symptoms of PTSD in the past month according to the DSM-5 (*Weathers et al., 2013*; *Blevins et al., 2015*; *Wortmann et al., 2016*; *Bovin et al., 2016*). Before answering the PCL-5, participants reported their

lifetime exposure to potentially traumatic events assessed with the Life Events Checklist for DSM-5 (LEC-5). Participants were asked to determine the worst traumatic event they had experienced and answer the PCL-5 according to the worst selected trauma. A clinical cut-off score of ≥33 was used to indicate probable PTSD (*Weathers et al., 2013*). The Pittsburgh Sleep Quality Index Addendum for PTSD (PSQI-A) is a seven-item valid questionnaire designed to assess the frequency of disruptive nocturnal behaviors common in individuals with PTSD (*Germain et al., 2005*). A clinical cut-off score of ≥4 was used to discriminate between participants with and without trauma-related sleep disturbances (*Germain et al., 2005*). Binge drinking was defined as having six or more units of alcohol on a single occasion (one unit corresponds to a single measure of spirits) at least once a month during the last year (*Bush et al., 1998*). The widely used Patient Health Questionnaire (PHQ-9) (*Kroenke et al., 2001*) and Generalized Anxiety Disorder Scale (GAD-7) (*Kroenke et al., 2010*; *Spitzer et al., 2006*; *Löwe et al., 2008*) were used to measure the presence of depression and anxiety symptoms, respectively, during the past 2 weeks. The standard cut-off score of ≥10 was used, indicating clinically relevant depression and anxiety (*Kroenke et al., 2001*; *Spitzer et al., 2006*).

Binary variables were created indicating whether an individual met the clinical cut-off for symptoms of each disorder above (0 = no, 1 = yes). The psychiatric resilience phenotype was derived by summing together the binary variables and reversing the score which resulted in a total score ranging from 0 to 5 where higher scores indicate greater psychiatric resilience in these trauma-exposed women. In addition, we created a binary variable where endorsement of 0 on all measures indicated high psychiatric resilience.

## Covariates

We summarised variables with a conceptual rationale for being associated both with ACEs and perceived coping ability or psychiatric resilience in *Appendix 1—figure 3*. We considered as covariates, age (at responding) and childhood deprivation (potential confounders), as well as educational level, employment status, civil status and current monthly income at responding (potential mediators). Age was divided into five groups for descriptive purposes: 18–29 years, 30–39 years, 40–49 years, 50–59 years, and 60 years and older. The age covariate was used as a continuous variable in all models. Education was categorized as primary education, secondary education (high school or vocational education), tertiary education A (BSc or equivalent), and tertiary education B (MSc or above). Civil status was divided into married or in a relationship and single or widowed, and employment status was divided into employed (including being a student and being on parental leave) and retired or on disability or sick leave. Current monthly income was categorized into the following groups: low income (<$2527), low-medium income ($2528-$4212), medium income ($4213-$5897), medium-high income ($5898-$8424), and high income (>$8425; conversion rates according to Central Bank of Iceland, October 17, 2018). Childhood deprivation was assessed with the question: Was your family's economic situation ever so bad that you suffered any deprivation as a consequence? For example, this could apply to deprivation of nutritious food and/or deprivation of warm clothes and appropriate footwear during the winter months, with response options ranging from 0 (never) to 4 (often). In addition, current perceived social support, measured with the Multidimensional Scale of Perceived Social Support (MSPSS) (*Zimet et al., 1988*), and perceived happiness, measured with a 10-point visual digital scale, were included in additional analyses (for further details, see the Appendix 1).

## Multiple imputation

The ACE-IQ scale and the five psychopathology scales (PHQ-9, GAD-7, PCL-5, PSQI-A and binge drinking) used to derive psychiatric resilience, had missing values which resulted in a reduced sample size (see *Appendix 1—figure 4*). We used multiple imputation (MI) to replace missing data with m = 20 rounds of imputations, using predictive mean matching (*van Buuren, 2019*). We imputed data for participants who responded to more than 75% of items on each scale and then calculated the total score for the scales. The subsequent analyses (described below) were conducted using the imputed dataset. For comparison, the main analyses were repeated in the original dataset with complete data.

## Statistical analyses

Descriptive characteristics were compared using Chi-square tests for categorical data and ANOVAs for continuous data.

Rank order correlations were used to determine (i) the correlation between perceived coping ability and psychiatric resilience, (ii) correlations between perceived coping ability and different measures of psychopathology (PHQ-9, GAD-7, PCL-5, PSQI-A, binge drinking) used to derive psychiatric resilience, and (iii) correlations between different ACE subtypes.

We used linear regression models assuming normally distributed errors and log-linear Poisson regression models with robust error variance to determine the associations between ACEs and perceived coping ability and psychiatric resilience, as continuous and binary outcomes (high perceived coping ability/ high psychiatric resilience), respectively. In all analyses, we adjusted for age and childhood deprivation (model 1) and then additionally for adult educational level, civil status, employment level and income (model 2).

We ran all models with ACE-IQ as a continuous predictor and as a categorical predictor where we compared resilience levels of unexposed women (0 ACEs) to resilience levels of those who had been exposed to increasing number of ACEs (1, 2, 3–4, and ≥5 ACEs). In addition, we carried out stratified analyses to assess whether the association between ACEs and resilience differed by levels of perceived social support (linear models). Furthermore, because parental separation/divorce is a common childhood experience, we carried out a sensitivity analysis excluding this item from the ACE-IQ score and re-ran the linear models. Finally, to preclude whether a particular response style influenced the observed associations between ACEs and resilience, we excluded women who scored low or high on happiness (10% top/bottom scores) and re-ran the linear models.

To determine the independent associations of specific types of ACEs with resilience, we ran analyses on perceived coping ability and psychiatric resilience for each of the 13 ACEs. We first examined each ACE type separately while adjusting for covariates, and then re-ran the analyses with all ACE subtypes entered simultaneously into the model, as ACEs frequently co-occur (*Radford et al., 2013*).

Standardized regression coefficients were reported from all linear regression analyses and prevalence ratios (PR) were reported from the Poisson regression analyses. All statistical analyses were performed using R (version 3.6.1).

## Results

### Characteristics of the sample

Descriptive statistics of the study population are presented in *Table 1*. Overall, 19.7% of participants reported no ACEs and 20.4% reported five or more ACEs. Middle aged women had on average higher scores on the ACE-IQ than younger or older women. Women with higher ACE scores were more often single or widowed, less educated, unemployed, had lower income and were more likely to report childhood deprivation and low perceived social support (*Table 1*).

### Bivariate associations

Both perceived coping ability and psychiatric resilience were consistently associated with older age, higher educational level, being in a relationship or married, being employed, higher income, lower childhood deprivation and higher perceived social support (*Table 2*). The two resilience measures were moderately correlated ($r_s$ = 0.47). Perceived coping ability was moderately negatively correlated with each of the measures used to derive the psychiatric resilience phenotype (i.e. symptoms of depression, anxiety, PTSD, and trauma-related sleep disturbances) but weakly correlated with binge drinking (*Appendix 1—table 2*). The 13 ACE subtypes were weakly to moderately correlated with each other ($r_s$ ranging from 0.06 to 0.46), with emotional abuse showing the strongest correlation with other ACEs (*Appendix 1—table 3*).

### Associations between ACEs and resilience

Linear models revealed that exposure to one or more ACEs (relative to 0 ACEs) was associated with lower perceived coping ability and psychiatric resilience in a dose-dependent manner (*Table 3*); every 1 SD unit increase in ACE-IQ scores was associated with lower levels of perceived coping ability ($\beta$ = −0.14; 95% CI −0.15,–0.13) and psychiatric resilience ($\beta$ = −0.28; 95% CI −0.29,–0.27) in the fully adjusted model. Associations between ACEs and perceived coping ability and psychiatric resilience were observed across levels of social support but were slightly stronger among women with low social support (*Appendix 1—table 4*). Sensitivity analyses showed that the associations remained evident

**Table 1.** Descriptive characteristics of the study population by number of adverse childhood experiences (ACE-IQ) (n = 26,198).

| | Number of ACEs | | | | | | | ACE-IQ sum score | |
|---|---|---|---|---|---|---|---|---|---|
| | Total | 0 ACE | 1 ACE | 2 ACEs | 3–4 ACEs | ≥ 5 ACEs | p-value global* | Mean (SD) | p-value global† |
| | N (%) | N (%) | N (%) | N (%) | N (%) | N (%) | | | |
| Total | 26,198 | 5,149 (19.7) | 5,567 (21.3) | 4,491 (17.1) | 5,640 (21.5) | 5,351 (20.4) | | 2.6 (2.4) | |
| Age, mean (SD) | 44.0 (13.6) | 43.7 (13.9) | 44.7 (13.9) | 44.6 (13.6) | 44.1 (13.4) | 43.1 (12.8) | < 0.001 | | |
| **Age groups** | | | | | | | | | |
| 18–29 years | 4,881 (18.6) | 1,043 (20.3) | 993 (17.8) | 805 (17.9) | 1,045 (18.5) | 995 (18.6) | < 0.001 | 2.6 (2.5) | < 0.001 |
| 30–39 years | 5,309 (20.3) | 1,074 (20.9) | 1,128 (20.3) | 864 (19.2) | 1,107 (19.6) | 1,136 (21.2) | | 2.7 (2.4) | |
| 40–49 years | 5,923 (22.6) | 1,092 (21.2) | 1,167 (21.0) | 1,003 (22.3) | 1,301 (23.1) | 1,360 (25.4) | | 2.8 (2.5) | |
| 50–59 years | 6,055 (23.1) | 1,091 (21.2) | 1,281 (23.0) | 1,088 (24.2) | 1,356 (24.0) | 1,239 (23.2) | | 2.7 (2.4) | |
| ≥ 60 years | 4,030 (15.4) | 849 (16.5) | 998 (17.9) | 731 (16.3) | 831 (14.7) | 621 (11.6) | | 2.3 (2.2) | |
| **Educational level** | | | | | | | | | |
| Primary education | 3,739 (14.3) | 442 (8.6) | 662 (11.9) | 602 (13.4) | 889 (15.8) | 1,144 (21.4) | < 0.001 | 3.4 (2.61) | < 0.001 |
| Secondary education | 8,013 (30.6) | 1,402 (27.2) | 1,658 (29.8) | 1,360 (30.3) | 1,744 (30.9) | 1,849 (34.6) | | 2.8 (2.32) | |
| Tertiary A (BSc or equivalent) | 8,359 (31.9) | 1,856 (36.0) | 1,872 (33.6) | 1,488 (33.1) | 1,782 (31.6) | 1,361 (25.4) | | 2.3 (2.09) | |
| Tertiary B (MSc or above) | 5,990 (22.9) | 1,437 (27.9) | 1,360 (24.4) | 1,022 (22.8) | 1,210 (21.5) | 961 (18.0) | | 2.3 (2.08) | |
| Unknown | 97 (0.4) | 12 (0.2) | 15 (0.3) | 19 (0.4) | 15 (0.3) | 36 (0.7) | | 3.7 (2.59) | |
| **Civil status** | | | | | | | | | |
| Married/in a relationship | 19,750 (75.4) | 4,061 (78.9) | 4,309 (77.4) | 3,442 (76.6) | 4,173 (74.0) | 3,765 (70.4) | < 0.001 | 2.5 (2.3) | < 0.001 |
| Single/widowed | 6,314 (24.1) | 1,070 (20.8) | 1,241 (22.3) | 1,030 (22.9) | 1,432 (25.4) | 1,541 (28.8) | | 2.9 (2.5) | |
| Unknown | 134 (0.5) | 18 (0.3) | 17 (0.3) | 19 (0.4) | 35 (0.6) | 45 (0.8) | | 3.6 (2.8) | |
| **Employment status** | | | | | | | | | |
| Employed/studying | 22,088 (84.3) | 4,639 (90.1) | 4,826 (86.7) | 3,888 (86.6) | 4,734 (83.9) | 4,001 (74.8) | < 0.001 | 2.5 (2.3) | < 0.001 |
| Retired/disability/sick leave | 3,941 (15.0) | 494 (9.6) | 718 (12.9) | 573 (12.8) | 854 (15.1) | 1,302 (24.3) | | 3.5 (2.8) | |
| Unknown | 169 (0.6) | 16 (0.3) | 23 (0.4) | 30 (0.7) | 52 (0.9) | 48 (0.9) | | 3.5 (2.6) | |
| **Income** | | | | | | | | | |
| Low income | 7,723 (29.5) | 1,206 (23.4) | 1,532 (27.5) | 1,216 (27.1) | 1,750 (31.0) | 2019 (37.7) | < 0.001 | 3.0 (2.6) | < 0.001 |
| Low-medium income | 7,862 (30.0) | 1,478 (28.7) | 1,663 (29.9) | 1,406 (31.3) | 1,717 (30.4) | 1,598 (29.9) | | 2.6 (2.4) | |
| Medium income | 6,050 (23.1) | 1,352 (26.3) | 1,363 (24.5) | 1,081 (24.1) | 1,236 (21.9) | 1,018 (19.0) | | 2.3 (2.2) | |
| High-medium income | 2,636 (10.1) | 654 (12.7) | 567 (10.2) | 466 (10.4) | 534 (9.5) | 415 (7.8) | | 2.2 (2.2) | |
| High income | 929 (3.5) | 250 (4.9) | 226 (4.1) | 141 (3.1) | 196 (3.5) | 116 (2.2) | | 2.0 (2.0) | |
| Unknown | 998 (3.8) | 209 (4.1) | 216 (3.9) | 181 (4.0) | 207 (3.7) | 185 (3.5) | | 2.5 (2.4) | |
| **Childhood deprivation** | | | | | | | | | |
| Never | 19,727 (75.3) | 4,843 (94.1) | 4,919 (88.4) | 3,628 (80.8) | 3,954 (70.1) | 2,383 (44.5) | < 0.001 | 2.0 (2.0) | < 0.001 |
| Rarely | 2,929 (11.2) | 218 (4.2) | 430 (7.7) | 516 (11.5) | 833 (14.8) | 932 (17.4) | | 3.5 (2.4) | |
| Sometimes | 2,320 (8.9) | 71 (1.4) | 193 (3.5) | 278 (6.2) | 627 (11.1) | 1,151 (21.5) | | 4.6 (2.5) | |
| Often | 1,169 (4.5) | 15 (0.3) | 21 (0.4) | 60 (1.3) | 206 (3.7) | 867 (16.2) | | 6.3 (2.6) | |
| Unknown | 53 (0.2) | 2 (0.0) | 4 (0.1) | 9 (0.2) | 20 (0.4) | 18 (0.3) | | 3.8 (2.0) | |
| **Perceived social support** | | | | | | | | | |
| Low | 6,332 (24.2) | 747 (14.5) | 995 (17.9) | 972 (21.6) | 1,582 (28.0) | 2036 (38.0) | < 0.001 | 3.5 (2.7) | < 0.001 |
| Moderate | 12,831 (40.0) | 2,487 (48.3) | 2,833 (50.9) | 2,282 (50.8) | 2,762 (49.0) | 2,467 (46.1) | | 2.5 (2.3) | |
| High | 6,151 (23.5) | 1,790 (34.8) | 1,579 (28.4) | 1,102 (24.5) | 1,075 (19.1) | 605 (11.3) | | 1.8 (1.9) | |
| Unknown | 884 (3.4) | 125 (2.4) | 160 (2.9) | 135 (3.0) | 221 (3.9) | 243 (4.5) | | 3.2 (2.5) | |

*p-values were obtained by $\chi^2$ tests, except for mean age which was compared with an ANOVA.

†p-values were obtained by ANOVA.

**Table 2.** Distribution of perceived coping ability (CD-RISC) and psychiatric resilience scores by sociodemographic characteristics.

| | Perceived coping ability | | Psychiatric resilience | |
|---|---|---|---|---|
| | Mean (SD) | p-value global* | Mean (SD) | p-value global* |
| Total | 28.0 (7.5) | | 3.6 (1.5) | |
| **Age groups** | | | | |
| 18–29 years | 24.8 (7.9) | < 0.001 | 3.1 (1.6) | < 0.001 |
| 30–39 years | 26.8 (7.5) | | 3.5 (1.6) | |
| 40–49 years | 27.8 (7.4) | | 3.7 (1.5) | |
| 50–59 years | 28.5 (7.2) | | 3.8 (1.4) | |
| ≥ 60 years | 28.8 (6.9) | | 4.0 (1.3) | |
| **Educational level** | | | | |
| Primary education | 24.2 (8.3) | < 0.001 | 3.1 (1.6) | < 0.001 |
| Secondary education | 26.3 (7.6) | | 3.4 (1.5) | |
| Tertiary A (BSc or equivalent) | 28.0 (7.0) | | 3.8 (1.4) | |
| Tertiary B (MSc or above) | 29.8 (6.5) | | 4.0 (1.3) | |
| Unknown | 23.6 (8.1) | | 3.1 (1.7) | |
| **Civil status** | | | | |
| Married/in a relationship | 27.7 (7.4) | < 0.001 | 3.7 (1.4) | < 0.001 |
| Single/widowed | 26.3 (7.9) | | 3.3 (1.6) | |
| Unknown | 24.7 (8.5) | | 2.7 (1.8) | |
| **Employment status** | | | | |
| Employed/studying | 28.0 (7.2) | < 0.001 | 3.7 (1.4) | < 0.001 |
| Retired/disability/sick leave | 23.9 (8.5) | | 3.0 (1.6) | |
| Unknown | 23.3 (9.0) | | 2.8 (1.6) | |
| **Income** | | | | |
| Low income | 24.5 (8.0) | < 0.001 | 3.1 (1.6) | < 0.001 |
| Low-medium income | 27.0 (7.2) | | 3.6 (1.5) | |
| Medium income | 29.3 (6.5) | | 4.0 (1.3) | |
| High-medium income | 30.9 (6.1) | | 4.1 (1.2) | |
| High income | 31.9 (6.1) | | 4.1 (1.2) | |
| Unknown | 26.6 (7.9) | | 3.7 (1.5) | |
| **Childhood deprivation** | | | | |
| Never | 27.9 (7.3) | < 0.001 | 3.8 (1.4) | < 0.001 |
| Rarely | 26.4 (7.6) | | 3.3 (1.6) | |
| Sometimes | 25.4 (7.9) | | 3.1 (1.6) | |
| Often | 25.0 (8.4) | | 2.6 (1.6) | |
| Unknown | 24.0 (8.9) | | 2.6 (1.6) | |
| **Perceived social support** | | | | |
| Low | 24.5 (8.0) | < 0.001 | 3.1 (1.6) | < 0.001 |
| Moderate | 27.2 (7.1) | | 3.7 (1.5) | |
| High | 30.9 (6.4) | | 4.1 (1.2) | |
| Unknown | 24.7 (8.0) | | 3.2 (1.6) | |

*p-values were obtained by ANOVAs.

**Table 3.** Associations between the number of ACEs and perceived coping ability (CD-RISC) and psychiatric resilience ($\beta$ and 95% CI)*.

| | | Perceived coping ability | | Psychiatric resilience | |
|---|---|---|---|---|---|
| | N (%) | Model 1† | Model 2 ‡ | Model 1† | Model 2 ‡ |
| ACE-IQ total score ** | 26,198 | –0.19 (-0.20,–0.17) | –0.14 (-0.15,–0.13) | –0.32 (-0.34,–0.31) | –0.28 (-0.29,–0.27) |
| Number of ACEs | | | | | |
| 0 ACE | 5,149 (19.7) | 0 (ref.) | 0 (ref.) | 0 (ref.) | 0 (ref.) |
| 1 ACE | 5,567 (21.3) | –0.07 (-0.08,–0.05) | –0.05 (-0.07,–0.04) | –0.09 (-0.10,–0.07) | –0.07 (-0.09,–0.06) |
| 2 ACE | 4,491 (17.1) | –0.10 (-0.11,–0.08) | –0.08 (-0.10,–0.07) | –0.13 (-0.15,–0.12) | –0.12 (-0.13,–0.10) |
| 3–4 ACE | 5,640 (21.5) | –0.16 (-0.17,–0.14) | –0.13 (-0.14,–0.11) | –0.22 (-0.23,–0.20) | –0.19 (-0.21,–0.18) |
| ≥ 5 ACEs | 5,351 (20.4) | –0.22 (-0.24,–0.20) | –0.16 (-0.18,–0.15) | –0.36 (-0.37,–0.34) | –0.31 (-0.33,–0.30) |

*Coefficients are standardized; **per 1 SD unit increase in ACE-IQ scores.

†adjusted for age and childhood deprivation.

‡additionally adjusted for education level, civil status, employment status and income.

when excluding parental separation/divorce from the ACE-IQ score (*Appendix 1—table 5*), and when excluding women with top/bottom 10% happiness values (*Appendix 1—table 6*).

Poisson models revealed that compared to women with 0 ACEs, women with ≥5 ACEs had a lower prevalence of high perceived coping ability (PR = 0.64, 95% CI 0.59, 0.70), and high psychiatric resilience (PR = 0.42, 95% CI 0.39, 0.45) in the fully adjusted model (*Table 4*). Every unit increase in the ACE-IQ scores was associated with lower prevalence of high perceived coping ability (PR = 0.93, 95% CI 0.92, 0.94) and high psychiatric resilience (PR = 0.87, 95% CI 0.86, 0.87).

All ACE subtypes were associated with lower levels of perceived coping ability and psychiatric resilience (*Appendix 1—figure 5*) as well as lower prevalence of high psychiatric resilience and high perceived coping ability except for physical neglect- and abuse, parental death or separation/divorce, incarceration of a household member, and community- and collective violence (*Appendix 1—figure 6*). After mutual adjustment for all ACE subtypes, we found that emotional neglect, being bullied, sexual abuse, and growing up with a mentally ill household member had consistent associations with both resilience measures, both in linear models (*Figure 1*) and Poisson models (*Figure 2*). Associations were also suggested for emotional abuse, domestic- and community violence with psychiatric resilience in the full model while associations for other ACEs were substantially attenuated (*Figures 1 and 2*).

**Table 4.** Prevalence Ratios (with 95% CI) of high perceived coping ability (CD-RISC ≥35) and high psychiatric resilience (absence of psychiatric morbidity) in relation to the number of ACEs.

| | | Perceived coping ability | | Psychiatric resilience | |
|---|---|---|---|---|---|
| | N (%) | Model 1†ᵃ | Model 2 ‡ | Model 1† | Model 2 ‡ |
| ACE-IQ total score* | 26,198 | 0.91 (0.90, 0.92) | 0.93 (0.92, 0.94) | 0.85 (0.84, 0.86) | 0.87 (0.86, 0.87) |
| Number of ACEs | | | | | |
| 0 ACE | 5,149 (19.7) | 1.00 (ref.) | 1.00 (ref.) | 1.00 (ref.) | 1.00 (ref.) |
| 1 ACE | 5,567 (21.3) | 0.83 (0.77, 0.89) | 0.87 (0.81, 0.93) | 0.82 (0.79, 0.84) | 0.84 (0.81, 0.87) |
| 2 ACE | 4,491 (17.1) | 0.71 (0.66, 0.77) | 0.75 (0.70, 0.81) | 0.73 (0.70, 0.76) | 0.75 (0.72, 0.78) |
| 3–4 ACE | 5,640 (21.5) | 0.61 (0.56, 0.66) | 0.66 (0.61, 0.71) | 0.59 (0.56, 0.61) | 0.62 (0.60, 0.65) |
| ≥ 5 ACEs | 5,351 (20.4) | 0.56 (0.51, 0.61) | 0.64 (0.59, 0.70) | 0.38 (0.35, 0.40) | 0.42 (0.39, 0.45) |

*per 1 SD unit increase in ACE-IQ scores.

†adjusted for age and childhood deprivation.

‡additionally adjusted for education level, civil status, employment status and income.

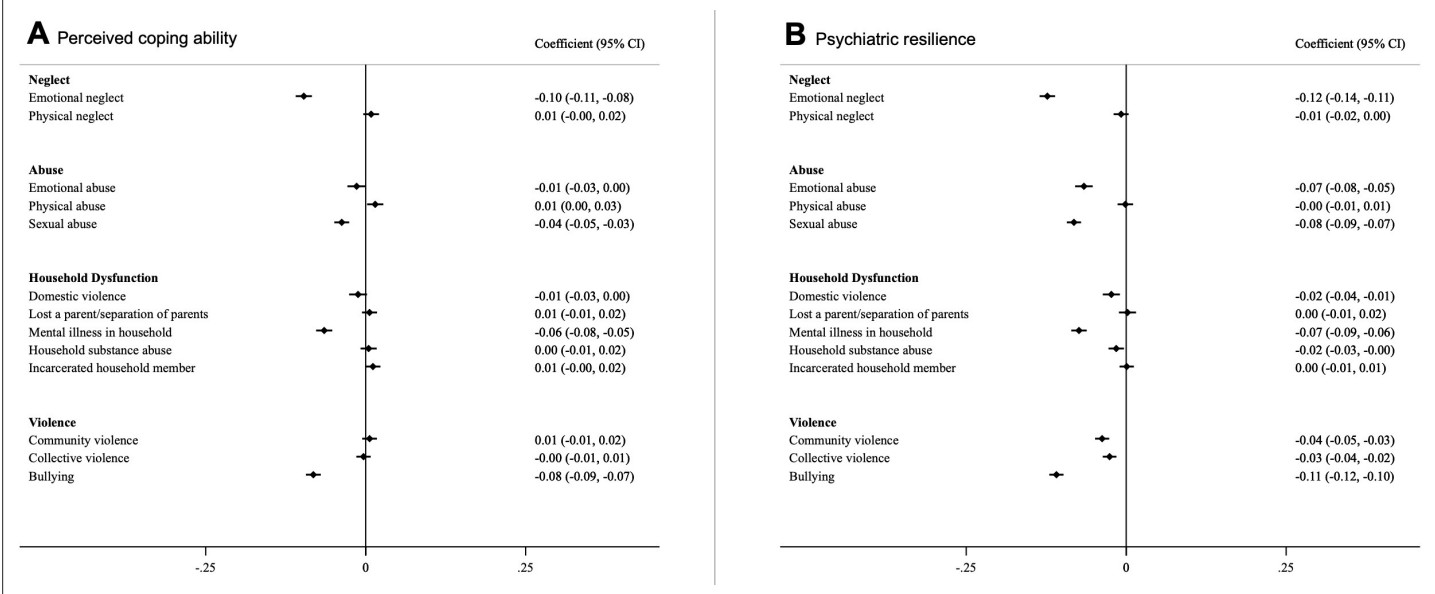

**Figure 1.** Associations between different types of ACEs and perceived coping ability (**A**) and psychiatric resilience (**B**) (β and 95% CI). Models were corrected for age, childhood deprivation, educational level, civil status, employment status, income and mutually adjusted for other ACEs. *Coefficients are standardized.

Overall, the results of the complete case analyses were similar to the results using multiple imputation (main analyses) in terms of both effect sizes and confidence intervals. See *Appendix 1—tables 7 and 8* (number of ACEs), and *Appendix 1—figures 7 and 8* (ACE subtypes) for the complete case analyses.

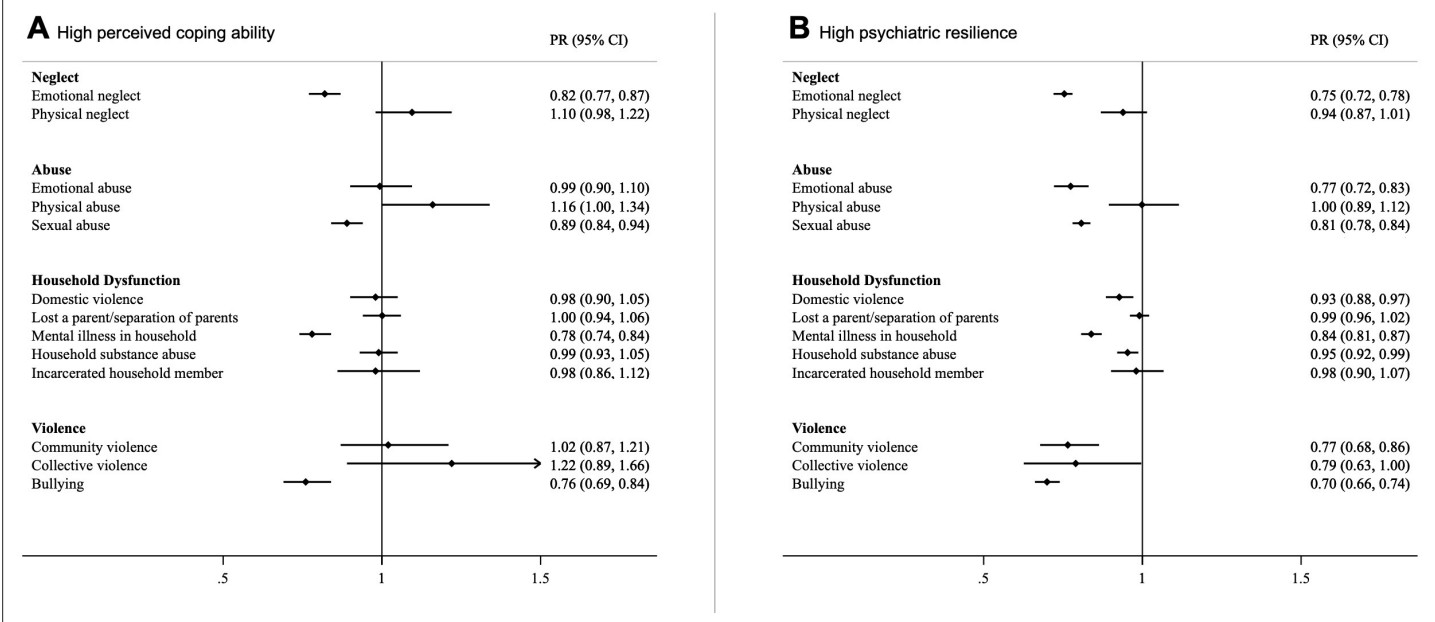

**Figure 2.** Prevalence Ratios (with 95% CI) of high perceived coping ability (**A**) and high psychiatric resilience (**B**) in relation to individual ACEs. Models were corrected for age, childhood deprivation, educational level, civil status, employment status, income and mutually adjusted for other ACEs.

## Discussion

In a large nationwide study of Icelandic women, a comprehensive measure of 13 adverse childhood experiences (ACEs) was negatively associated with two distinct measures of resilience in adulthood in a dose-dependent manner. Indeed, women who endorsed five or more ACEs had 36% lower prevalence of high perceived coping ability and 58% lower prevalence of high psychiatric resilience compared to women who endorsed 0 ACEs. In addition, specific ACEs including emotional neglect, bullying, sexual abuse and growing up with a mentally ill household member were strongly associated with lower adult resilience.

To our knowledge, the present study is the first to consider a wide range of ACEs in relation to adult resilience, and to address their association with two distinct resilience measures. Overall, our results are in line with previous studies examining the association between childhood maltreatment and adult resilience (*Mersky and Topitzes, 2010*; *Topitzes et al., 2013*; *McGloin and Widom, 2001*; *Williams et al., 2006*; *Campbell-Sills et al., 2009*; *Nishimi et al., 2020*). However, the existing evidence base is limited by relatively small sample sizes, consideration of only few ACEs (*McGloin and Widom, 2001*; *Williams et al., 2006*; *Campbell-Sills et al., 2009*; *Nishimi et al., 2020*), and/or the use of composite measures of childhood adversity (*Mersky and Topitzes, 2010*; *Topitzes et al., 2013*).

Our results indicating a dose-response relationship between the cumulative number of ACEs and lower adult resilience is consistent with a recent cross-sectional study by *Nishimi et al., 2020* who observed a similar pattern between four ACEs (emotional abuse, physical abuse, sexual abuse, and witnessing domestic violence) and perceived coping ability, as measured with the CD-RISC. We have now extended these findings to psychiatric resilience as an outcome, and further demonstrated this pattern for a broad set of ACEs in a large nationwide study representing 30% of the Icelandic female population. Furthermore, our results suggest that all ACE subtypes are associated with perceived coping ability and psychiatric resilience. However, when mutually adjusting for other ACEs, only emotional neglect, sexual abuse, being bullied and growing up with a mentally ill household member were consistently associated with both resilience measures.

In line with previous literature (*Nishimi et al., 2021*; *Sheerin et al., 2018*), we found that self-assessed coping ability and outcome-based psychiatric resilience were only moderately correlated with each other, indicating these may reflect different patterns of adaptive functioning following adversity. Collectively, our research adds to the growing literature on the nature of resilience as a construct. The associations between ACEs and adult resilience were independent of age, childhood deprivation and, importantly, other adult socio-demographic factors (i.e. educational level, civil status, employment level and income), suggesting that ACEs affect resilience over and above these potential mediating variables. Indeed, one possible mechanism through which ACEs could influence resilience, are functional outcomes in adulthood (e.g. social, financial and/or educational functioning). Previous research has found that ACEs are associated with greater risk of poor educational and financial outcomes, as well as poor social functioning (*Bellis et al., 2014*; *Copeland et al., 2018*), both adult factors that have previously been associated with resilience (*Nishimi et al., 2021*; *Campbell-Sills et al., 2009*; *Southwick et al., 2016*). However, in the current study, effect sizes only diminished slightly when we additionally adjusted for adult socio-demographic factors, which indicates that adult characteristics such as education and employment level do not compensate for the deleterious impact of ACEs on adult resilience. This suggests that adult resilience may largely be determined in childhood and that situational factors in adulthood (e.g. high social support) only marginally buffer the association between ACEs and adult resilience.

The main strengths of the study include the population-based design and large sample size of the SAGA cohort, which represents the Icelandic adult female population in terms of distribution of age, residence, education and income. The wealth of measures in the SAGA cohort baseline assessments made it possible to derive two types of resilience measures and examine a wide range of ACEs. The wealth of relevant data also allowed us to adjust for childhood deprivation, an important confounder when examining ACEs and adult outcomes (*Copeland et al., 2018*; *Arseneault et al., 2011*; *Bellis et al., 2017*), which has not been taken into account in previous ACE-resilience studies. However, our study also has several limitations. First, the cross-sectional nature of our data does not allow us to make any inferences about the directionality of the studied associations. Second, we cannot rule out that the observed association between ACE growing up with a mentally ill household member and resilience, is due to a genetic predisposition for psychopathology rather than the experience

itself. Future genetically informative studies will need to examine the extent to which this association is confounded by genetic factors. Although we adjusted for an array of important confounding factors, we cannot exclude the possibility that unmeasured or residual confounding may contribute to our results. In addition, ACEs were retrospectively reported and thus may be subject to recall bias. However, previous studies have shown acceptable validity for retrospective assessments of ACEs (*Reuben et al., 2016*; *Hardt and Rutter, 2004*; *Widom and Shepard, 1996*; *Widom and Morris, 1997*), although they may be influenced by current mental health status or response style (*Reuben et al., 2016*). Yet, the similar results obtained from our sensitivity analyses excluding individuals with extreme values on the happiness assessment, reduce concerns that our results are due to a particular response style. Finally, our results are based on an exclusively female sample, therefore, future studies should explore whether there are qualitative differences in how ACEs relate to adult resilience among men as well as among sexual and gender minorities.

In conclusion, in a large nationwide-representative female population, we observed a negative association between cumulative exposure of ACEs and two distinct measures of adult resilience. Specific ACEs, including exposure to emotional neglect and bullying, sexual abuse, and growing up with a mentally ill household member were consistently associated with lower adult resilience. If these findings are confirmed through prospective designs, there may be huge societal benefits of prevention strategies targeting the protection of children against traumatic occurrences and their consequences. Future research is needed to address how children exposed to ACEs can be supported to reduce risks of compromised adult resilience and health inequalities.

## Acknowledgements

Funding This work was supported by the European Research Council (UAV, grant number 726413), and the Icelandic Center for Research (Grant of excellence; UAV, grant number 163362–051). HBD was supported by a doctoral grant from the University of Iceland Research Fund. The funders had no role in study design, data collection and interpretation, or the decision to submit the work for publication.

## Additional information

### Funding

| Funder | Grant reference number | Author |
|---|---|---|
| Icelandic Centre for Research | Doctoral grant | Hilda Björk Daníelsdóttir |
| H2020 European Research Council | Consolidator grant grant number 726413 | Unnur Anna Valdimarsdóttir |
| Icelandic Centre for Research | Grant of excellence grant number 163362-051 | Unnur Anna Valdimarsdóttir |

The funders had no role in study design, data collection and interpretation, or the decision to submit the work for publication.

### Author contributions

Hilda Björk Daníelsdóttir, Conceptualization, Formal analysis, Methodology, Writing – original draft, Writing – review and editing; Thor Aspelund, Data curation, Formal analysis, Methodology, Supervision, Writing – review and editing; Edda Bjork Thordardottir, Thorhildur Halldorsdottir, Arna Hauksdóttir, Methodology, Project administration, Resources, Writing – review and editing; Katja Fall, Fang Fang, Qian Yang, Karmel W Choi, Beatrice Kennedy, Writing – review and editing, Methodology; Gunnar Tómasson, Huan Song, Methodology, Project administration, Writing – review and editing; Harpa Rúnarsdóttir, Project administration, Writing – review and editing; Donghao Lu, Data curation, Methodology, Writing – review and editing; Jóhanna Jakobsdóttir, Data curation, Project administration, Writing – review and editing; Unnur Anna Valdimarsdóttir, Conceptualization, Funding acquisition, Methodology, Resources, Supervision, Writing – original draft, Writing – review and editing

## Author ORCIDs

Hilda Björk Daníelsdóttir  http://orcid.org/0000-0002-4967-2495
Thor Aspelund  http://orcid.org/0000-0002-7998-5433
Edda Bjork Thordardottir  http://orcid.org/0000-0003-3775-9611
Katja Fall  http://orcid.org/0000-0002-3649-2639
Fang Fang  http://orcid.org/0000-0002-3310-6456
Gunnar Tómasson  http://orcid.org/0000-0002-1797-7091
Qian Yang  http://orcid.org/0000-0001-6389-3975
Karmel W Choi  http://orcid.org/0000-0002-3914-2431
Beatrice Kennedy  http://orcid.org/0000-0002-0066-4814
Thorhildur Halldorsdottir  http://orcid.org/0000-0003-0637-8912
Donghao Lu  http://orcid.org/0000-0002-4186-8661
Huan Song  http://orcid.org/0000-0003-3845-8079
Jóhanna Jakobsdóttir  http://orcid.org/0000-0002-8019-9683
Arna Hauksdóttir  http://orcid.org/0000-0002-4253-1059
Unnur Anna Valdimarsdóttir  http://orcid.org/0000-0001-5382-946X

## Ethics

Human subjects: The study was approved by the National Bioethics Committee (NBC number: 17-238) and all participants gave informed consent before participation.

## Decision letter and Author response

Decision letter https://doi.org/10.7554/eLife.71770.sa1
Author response https://doi.org/10.7554/eLife.71770.sa2

# Additional files

## Supplementary files

• Transparent reporting form
• Reporting standard 1. Strobe Checklist.
• Source code 1. R scripts for the main analyses.

## Data availability

The data used in this study are compiled in the Stress-And-Gene-Analysis (SAGA) cohort. We cannot make the data publicly available because of Icelandic laws regarding data protection and the approval for the current study granted by the National Bioethics Committee (NBC) of Iceland. The SAGA cohort contains extremely sensitive data and all use of data is restricted to scientific purposes only subjected to approval of the NBC (email: vsn@vsn.is). Interested researchers can obtain access to deidentified data by submitting a proposal to the SAGA cohort data management board (email: afallasaga@hi.is) which assists with submitting an amendment to the NBC. The corresponding author of the present study submitted a research proposal to the SAGA cohort data management board / the NBC and got access only to deidentified data, that cannot be shared in any way.

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

# Appendix 1

## Perceived coping ability

The CD-RISC total scores were divided into quintiles which resulted in 21.50% of the sample in the lowest quintile (raw CD-RISC scores = 0–21), 20.95% in the lower middle quintile (raw CD-RISC scores = 22–26), 20.63% in the middle quintile (raw CD-RISC scores = 27–30), 17.99% in the higher middle quintile (raw CD-RISC scores = 31–34) and 18.93% in the highest quintile (raw CD-RISC scores = 35–40).

## Covariate assessment

Social support was assessed with the Multidimensional Scale of Perceived Social Support (MSPSS) (*Zimet et al., 1988*). The instrument consists of 12 items answered on a 7-point scale ranging from 0 (very strongly disagree) to 6 (very strongly agree). Items were summed to create a total score ranging from 0 to 72, with higher scores indicating higher levels of perceived social support. In addition, the total scores were divided into quartiles which resulted in 25.09% of the sample in the lowest quartile (raw MSPSS scores = 0–49), 25.52% of the sample in the low-middle quartile (raw MSPSS scores = 50–61), 24.92% of the sample in the high-middle quartile (raw MSPSS scores = 62–69) and 24.47% of the sample in the highest quartile (raw MSPSS scores = 70–72). A categorical variable was then created where the highest tertile was used to define a high level of social support, the two middle quartiles were merged and used together to define a moderate level of social support and the lowest quartile was used to define a low level of social support. Happiness was assessed with the question "In general, how would you rate your happiness?", and participants rated their happiness with a slider ranging from 1 to 10.

**Appendix 1—table 1.** List of the 30 ACE-IQ items used to derive the 13 different ACEs and their response options.

| ACE item | Scoring* |
|---|---|
| *Neglect* | |
| **Emotional neglect** | |
| Did your parents/guardians understand your problems and worries? | Always = 0, Most of the time = 1, Sometimes = 2, Rarely = 3, Never = 4 |
| Did your parents/guardians really know what you were doing with your free time when you were not at school or work? | Always = 0, Most of the time = 1, Sometimes = 2, Rarely = 3, Never = 4 |
| **Physical neglect** | |
| How often did your parents/guardians not give you enough food even when they could easily have done so? | Never = 0, Once = 1, A few times = 2, Many times = 3 |
| Were your parents/guardians too drunk or intoxicated by drugs to take care of you? | Never = 0, Once = 1, A few times = 2, Many times = 3 |
| How often did your parents/guardians not send you to school even when it was available? | Never = 0, Once = 1, A few times = 2, Many times = 3 |
| *Abuse* | |
| **Emotional abuse** | |
| Did a parent, guardian or other household member yell, scream or swear at you, insult or humiliate you? | Never = 0, Once = 1, A few times = 2, Many times = 3 |
| Did a parent, guardian or other household member threaten to, or actually, abandon you or throw you out of the house? | Never = 0, Once = 1, A few times = 2, Many times = 3 |
| **Physical abuse** | |
| Did a parent, guardian or other household member spank, slap, kick, punch or beat you up? | Never = 0, Once = 1, A few times = 2, Many times = 3 |
| Did a parent, guardian or other household member hit or cut you with an object, such as a stick (or cane), bottle, club, knife, whip etc? | Never = 0, Once = 1, A few times = 2, Many times = 3 |
| **Sexual abuse** | |

*Appendix 1—table 1 Continued on next page*

*Appendix 1—table 1 Continued*

| ACE item | Scoring* |
|---|---|
| Did someone touch or fondle you in a sexual way when you did not want them to? | Never = 0, Once = 1, A few times = 2, Many times = 3 |
| Did someone make you touch their body in a sexual way when you did not want them to? | Never = 0, Once = 1, A few times = 2, Many times = 3 |
| Did someone attempt oral, anal, or vaginal intercourse with you when you did not want them to? | Never = 0, Once = 1, A few times = 2, Many times = 3 |
| Did someone actually have oral, anal, or vaginal intercourse with you when you did not want them to? | Never = 0, Once = 1, A few times = 2, Many times = 3 |
| *Household dysfunction* | |
| Domestic violence | |
| Did you see or hear a parent or household member in your home being yelled at, screamed at, sworn at, insulted or humiliated? | Never = 0, Once = 1, A few times = 2, Many times = 3 |
| Did you see or hear a parent or household member in your home being slapped, kicked, punched or beaten up? | Never = 0, Once = 1, A few times = 2, Many times = 3 |
| Did you see or hear a parent or household member in your home being hit or cut with an object, such as a stick (or cane), bottle, club, knife, whip etc.? | Never = 0, Once = 1, A few times = 2, Many times = 3 |
| Lost a parent / separation of parents | |
| Were your parents ever separated or divorced? | No = 0, Yes = 1 |
| Did your mother, father or guardian die? | No = 0, Yes = 1 |
| Mental illness in household | |
| Did you live with a household member who was depressed, mentally ill or suicidal? | No = 0, Yes = 1 |
| Household substance abuse | |
| Did you live with a household member who was a problem drinker or alcoholic, or misused street or prescription drugs? | No = 0, Yes = 1 |
| Incarcerated household member | |
| Did you live with a household member who was ever sent to jail or prison? | No = 0, Yes = 1 |
| *Other violence* | |
| Community violence | |
| Did you see or hear someone being beaten up in real life? | Never = 0, Once = 1, A few times = 2, Many times = 3 |
| Did you see or hear someone being stabbed or shot in real life? | Never = 0, Once = 1, A few times = 2, Many times = 3 |
| Did you see or hear someone being threatened with a knife or gun in real life? | Never = 0, Once = 1, A few times = 2, Many times = 3 |
| Collective violence | |
| During the first 18 years of your life, were you exposed to war/ collective violence (e.g. from gangs or police)?† | No = 0, Yes = 1 |
| Were you forced to go and live in another place due to any of these events? | Never = 0, Once = 1, A few times = 2, Many times = 3 |
| Did you experience the deliberate destruction of your home due to any of these events? | Never = 0, Once = 1, A few times = 2, Many times = 3 |
| Were you beaten up by soldiers, police, militia, or gangs? | Never = 0, Once = 1, A few times = 2, Many times = 3 |
| Was a family member or friend killed or beaten up by soldiers, police, militia, or gangs? | Never = 0, Once = 1, A few times = 2, Many times = 3 |
| Bullying | |
| How often were you bullied? | Never = 0, Once = 1, A few times = 2, Many times = 3 |

*all items also had the option „can't/don't want to answer".
†this is a screening question, only participants that responded yes got the following four questions.

**Appendix 1—table 2.** Rank order correlations for perceived coping (CD-RISC) and different measures of psychopathology used to derive the psychiatric resilience phenotype (n = 26,198).

|  | CD-RISC | PHQ-9 | GAD-7 | PCL-5 | PSQI-A | Binge drinking |
|---|---|---|---|---|---|---|
| CD-RISC | 1 |  |  |  |  |  |
| PHQ-9 | −0.55 | 1 |  |  |  |  |
| GAD-7 | −0.51 | 0.76 | 1 |  |  |  |
| PCL-5 | −0.48 | 0.70 | 0.66 | 1 |  |  |
| PSQI-A | −0.41 | 0.62 | 0.62 | 0.63 | 1 |  |
| Binge drinking | −0.04 | 0.10 | 0.09 | 0.07 | 0.09 | 1 |

**Appendix 1—table 3.** Rank order correlations for ACE subtypes (n = 26,198).

|  | Emotional abuse | Physical abuse | Sexual abuse | Emotional neglect | Physical neglect | Domestic violence | Lost a parent/ separation | Mental illness in household | Household substance abuse | Incarcerated household member | Community violence | Collective violence | Bullying |
|---|---|---|---|---|---|---|---|---|---|---|---|---|---|
| Emotional abuse | 1 |  |  |  |  |  |  |  |  |  |  |  |  |
| Physical abuse | 0.46 | 1 |  |  |  |  |  |  |  |  |  |  |  |
| Sexual abuse | 0.19 | 0.14 | 1 |  |  |  |  |  |  |  |  |  |  |
| Emotional neglect | 0.36 | 0.23 | 0.22 | 1 |  |  |  |  |  |  |  |  |  |
| Physical neglect | 0.29 | 0.18 | 0.14 | 0.27 | 1 |  |  |  |  |  |  |  |  |
| Domestic violence | 0.51 | 0.31 | 0.18 | 0.34 | 0.30 | 1 |  |  |  |  |  |  |  |
| Lost a parent/ separation of parents | 0.16 | 0.09 | 0.12 | 0.19 | 0.19 | 0.21 | 1 |  |  |  |  |  |  |
| Mental illness in household | 0.33 | 0.18 | 0.14 | 0.26 | 0.23 | 0.37 | 0.19 | 1 |  |  |  |  |  |
| Household substance abuse | 0.20 | 0.11 | 0.16 | 0.24 | 0.29 | 0.36 | 0.26 | 0.31 | 1 |  |  |  |  |
| Incarcerated household member | 0.14 | 0.11 | 0.09 | 0.12 | 0.18 | 0.22 | 0.14 | 0.18 | 0.25 | 1 |  |  |  |
| Community violence | 0.20 | 0.20 | 0.10 | 0.13 | 0.14 | 0.18 | 0.08 | 0.12 | 0.11 | 0.11 | 1 |  |  |
| Collective violence | 0.09 | 0.08 | 0.06 | 0.07 | 0.08 | 0.09 | 0.06 | 0.07 | 0.06 | 0.10 | 0.11 | 1 |  |
| Bullying | 0.21 | 0.16 | 0.13 | 0.15 | 0.11 | 0.15 | 0.06 | 0.16 | 0.08 | 0.06 | 0.12 | 0.08 | 1 |

**Appendix 1—table 4.** Associations between the number of ACEs and perceived coping ability (CD-RISC) and psychiatric resilience stratified by social support (n = 25,314) (β and 95% CI).

|  |  | Perceived coping ability | | | Psychiatric resilience | | |
|---|---|---|---|---|---|---|---|
|  | N (%) | Low support | Moderate support | High support | Low support | Moderate support | High support |
| Number of ACEs[*] |  |  |  |  |  |  |  |
| 0 ACE | 5,024 (19.8) | 0 (ref.) | 0 (ref.) | 0 (ref.) | 0 (ref.) | 0 (ref.) | 0 (ref.) |
| 1 ACE | 5,407 (21.3) | −0.06 (−0.10,−0.02) | −0.05 (−0.06,−0.03) | −0.02 (−0.05,−0.00) | −0.10 (−0.14,−0.06) | −0.07 (−0.09,−0.05) | −0.05 (−0.07,−0.03) |
| 2 ACE | 4,356 (17.2) | −0.08 (−0.12,−0.05) | −0.06 (−0.08,−0.04) | −0.06 (−0.08,−0.03) | −0.13 (−0.16,−0.09) | −0.11 (−0.13,−0.09) | −0.09 (−0.11,−0.07) |
| 3–4 ACE | 5,419 (21.4) | −0.11 (−0.14,−0.07) | −0.09 (−0.11,−0.07) | −0.08 (−0.11,−0.06) | −0.22 (−0.25,−0.17) | −0.16 (−0.18,−0.14) | −0.13 (−0.16,−0.10) |
| ≥ 5 ACEs | 5,108 (20.2) | −0.14 (−0.17,−0.10) | −0.10 (−0.12,−0.08) | −0.08 (−0.11,−0.05) | −0.32 (−0.36,−0.28) | −0.27 (−0.29,−0.25) | −0.21 (−0.24,−0.1) |

[*]Coefficients are standardized; adjusted for age, childhood deprivation, education level, civil status, employment status and income.

**Appendix 1—table 5.** Associations between the number of ACEs (excluding parental divorce/separation) and perceived coping ability (CD-RISC) and psychiatric resilience (β and 95% CI)*.

| | N (%) | Perceived coping ability | | Psychiatric resilience | |
|---|---|---|---|---|---|
| | | Model 1[a] | Model 2[b] | Model 1[a] | Model 2[b] |
| Number of ACEs* | | | | | |
| 0 ACE | 6,095 (23.3) | 0 (ref.) | 0 (ref.) | 0 (ref.) | 0 (ref.) |
| 1 ACE | 6,201 (23.7) | –0.08 (–0.09,–0.06) | –0.06 (–0.08,–0.05) | –0.10 (–0.11,–0.08) | –0.08 (–0.10,–0.07) |
| 2 ACE | 4,877 (18.6) | –0.12 (–0.13,–0.10) | –0.10 (–0.11,–0.08) | –0.16 (–0.17,–0.14) | –0.14 (–0.16,–0.13) |
| 3–4 ACE | 5,582 (21.3) | –0.18 (–0.19,–0.16) | –0.14 (–0.16,–0.13) | –0.25 (–0.27,–0.24) | –0.23 (–0.24,–0.21) |
| ≥ 5 ACEs | 3,443 (13.1) | –0.20 (–0.21,–0.19) | –0.15 (–0.15,–0.12) | –0.33 (–0.35,–0.32) | –0.29 (–0.30,–0.28) |

*Coefficients are standardized; [a]adjusted for age and childhood deprivation; [b]additionally adjusted for education level, civil status, employment status and income.

**Appendix 1—table 6.** Associations between the number of ACEs and perceived coping ability (CD-RISC) and psychiatric resilience excluding participants with ≈10% lowest and highest happiness values (raw scores 1–5 and 10) (n = 15,449) (β and 95% CI)*.

| | N (%) | Perceived coping ability | | Psychiatric resilience | |
|---|---|---|---|---|---|
| | | Model 1[a] | Model 2[b] | Model 1[a] | Model 2[b] |
| Number of ACEs* | | | | | |
| 0 ACE | 4,088 (20.38) | 0 (ref.) | 0 (ref.) | 0 (ref.) | 0 (ref.) |
| 1 ACE | 4,384 (21.86) | –0.05 (–0.07,–0.04) | –0.04 (–0.06,–0.02) | –0.09 (–0.10,–0.07) | –0.08 (–0.09,–0.06) |
| 2 ACE | 3,573 (17.82) | –0.08 (–0.09,–0.06) | –0.06 (–0.08,–0.05) | –0.12 (–0.14,–0.11) | –0.11 (–0.13,–0.10) |
| 3–4 ACE | 4,341 (21.65) | –0.12 (–0.14,–0.10) | –0.10 (–0.12,–0.08) | –0.19 (–0.20,–0.17) | –0.17 (–0.19,–0.16) |
| ≥ 5 ACEs | 3,669 (18.29) | –0.15 (–0.17,–0.13) | –0.12 (–0.13,–0.10) | –0.31 (–0.33,–0.29) | –0.28 (–0.30,–0.27) |

*Coefficients are standardized; [a]adjusted for age and childhood deprivation; [b]additionally adjusted for education level, civil status, employment status and income.

**Appendix 1—table 7.** Associations between the number of ACEs and perceived coping ability (CD-RISC) and psychiatric resilience (β and 95% CI)*, complete case analyses.

| | N (%) | Perceived coping ability | | Psychiatric resilience | |
|---|---|---|---|---|---|
| | | Model 1[a] | Model 2[b] | Model 1[a] | Model 2[b] |
| ACE-IQ total score* | 19,637 | –0.18 (-0.19,–0.16) | –0.13 (-0.15,–0.12) | –0.31 (-0.32,–0.30) | –0.28 (-0.29,–0.26) |
| Number of ACEs | | | | | |
| 0 ACE | 4,377 (22.3) | 0 (ref.) | 0 (ref.) | 0 (ref.) | 0 (ref.) |
| 1 ACE | 4,496 (22.9) | –0.07 (-0.09,–0.05) | –0.06 (-0.07,–0.04) | –0.08 (-0.10,–0.06) | –0.07 (-0.09,–0.05) |
| 2 ACE | 3,437 (17.5) | –0.10 (-0.11,–0.08) | –0.08 (-0.10,–0.06) | –0.13 (-0.14,–0.11) | –0.11 (-0.13,–0.10) |
| 3–4 ACE | 3,985 (20.3) | –0.14 (-0.16,–0.13) | –0.12 (-0.14,–0.10) | –0.20 (-0.21,–0.18) | –0.18 (-0.19,–0.16) |
| ≥ 5 ACEs | 3,342 (17.0) | –0.20 (-0.22,–0.18) | –0.15 (-0.17,–0.14) | –0.33 (-0.35,–0.31) | –0.29 (-0.31,–0.27) |

*Coefficients are standardized; **per 1 SD unit increase in ACE-IQ scores; [a]adjusted for age and childhood deprivation; [b]additionally adjusted for education level, civil status, employment status and income.

**Appendix 1—table 8.** Prevalence Ratios (with 95% CI) of high perceived coping ability (CD-RISC ≥35) and high psychiatric resilience (absence of psychiatric morbidity) in relation to the number of ACEs, complete case analyses.

|  |  | High perceived coping ability | | High psychiatric resilience | |
|---|---|---|---|---|---|
|  | N (%) | Model 1* | Model 2[b] | Model 1* | Model 2[b] |
| ACE-IQ total score* | 19,637 | 0.92 (0.90, 0.93) | 0.94 (0.92, 0.95) | 0.86 (0.85, 0.87) | 0.87 (0.87, 0.88) |
| Number of ACEs |  |  |  |  |  |
| 0 ACE | 4,377 (22.3) | 1.00 (ref.) | 1.00 (ref.) | 1.00 (ref.) | 1.00 (ref.) |
| 1 ACE | 4,496 (22.9) | 0.83 (0.77, 0.89) | 0.87 (0.81, 0.93) | 0.84 (0.81, 0.87) | 0.86 (0.83, 0.89) |
| 2 ACE | 3,437 (17.5) | 0.73 (0.67, 0.79) | 0.77 (0.71, 0.83) | 0.74 (0.71, 0.78) | 0.76 (0.73, 0.80) |
| 3–4 ACE | 3,985 (20.3) | 0.66 (0.60, 0.71) | 0.70 (0.65, 0.76) | 0.62 (0.59, 0.65) | 0.65 (0.62, 0.68) |
| ≥ 5 ACEs | 3,342 (17.0) | 0.59 (0.53, 0.65) | 0.67 (0.60, 0.74) | 0.41 (0.38, 0.44) | 0.44 (0.41, 0.48) |

*adjusted for age and childhood deprivation; [b]additionally adjusted for education level, civil status, employment status and income.

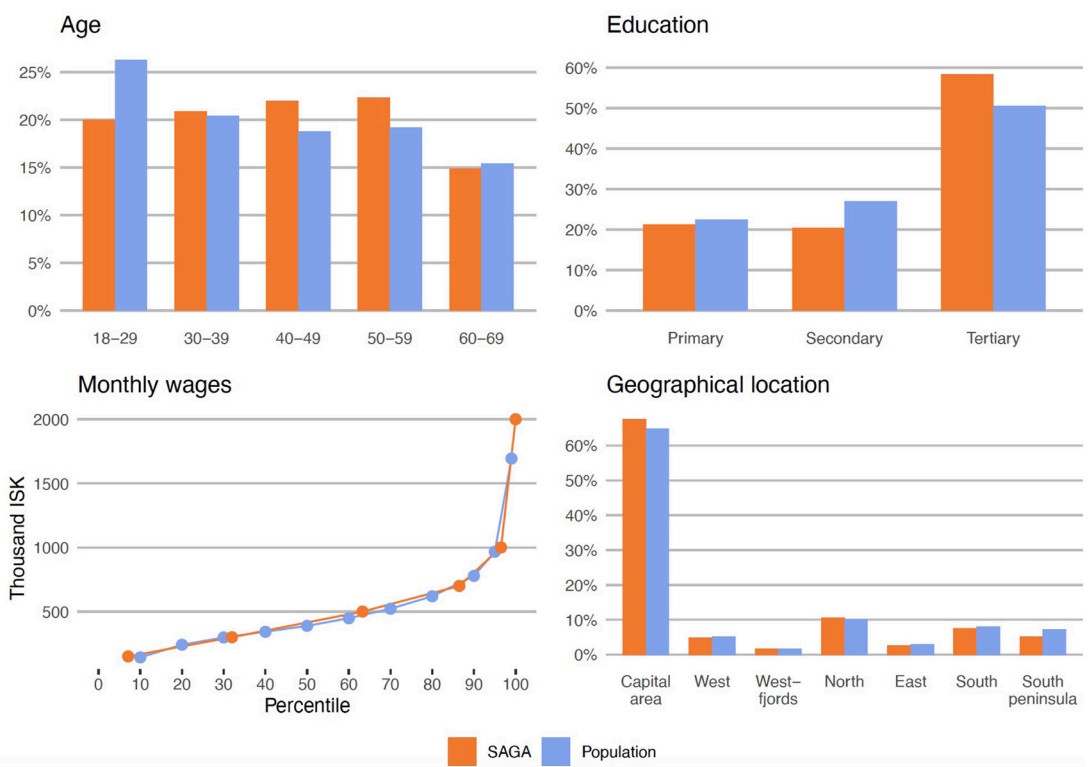

**Appendix 1—figure 1.** Sociodemographic characteristics of SAGA participants vs. the general female population of Iceland (see further: https://www.afallasaga.is/nidurstodur).

n = 30,403 women 18-69 years participated in the SAGA cohort

Women reporting no lifetime trauma (n = 2563)

n = 27,840

Missing information perceived coping ability (n = 509)

n = 27,331

More than 25% missing on psychopathology measures used to derive psychiatric resilience (n = 1040)

n = 26,291

More than 25% missing on ACE-IQ (n = 93)

Analytic sample n = 26,198

**Appendix 1—figure 2.** Flow-chart of the analytic sample.

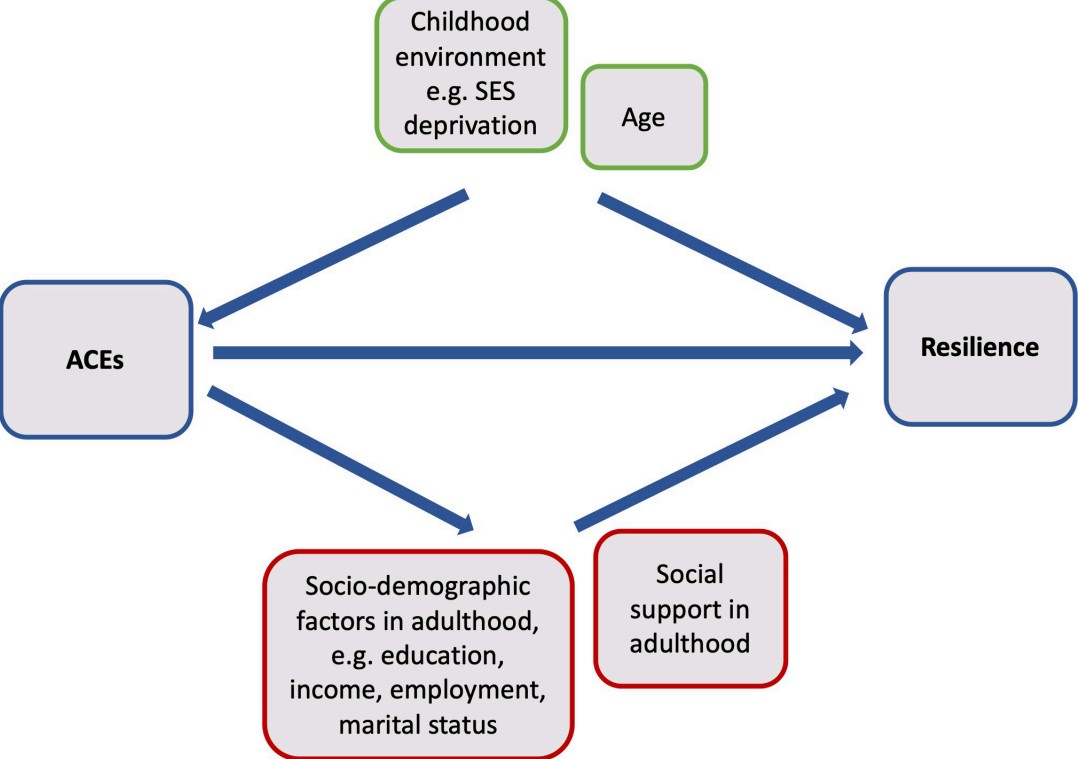

**Appendix 1—figure 3.** Proposed causal model with alternative pathways of how ACEs could influence resilience in adulthood.
Note: Boxes in green indicate potential confounders of the association between ACEs and adult resilience, whereas boxes in red indicate potential mediators of the association.

n = 30,403 women 18-69 years participated in the SAGA cohort

Women reporting no lifetime trauma (n = 2563)

n = 27,840

Missing information perceived coping ability (n = 509)

n = 27,331

Missing information psychiatric resilience (n = 4745)

n = 22,586

Missing information ACEs (n = 2949)

Analytic sample n = 19,637

**Appendix 1—figure 4.** Flow-chart of the analytic sample (complete case analysis).

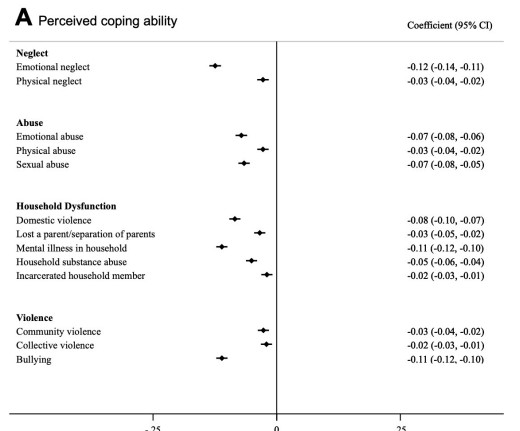
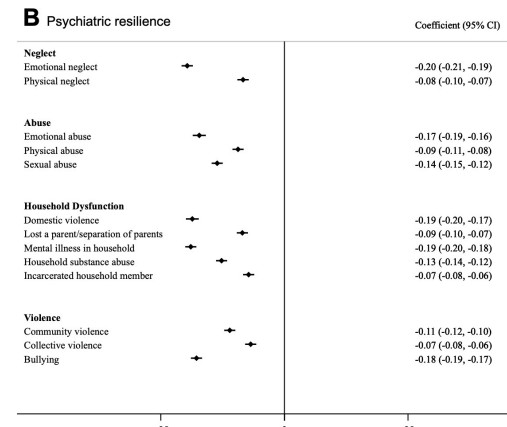

**Appendix 1—figure 5.** Associations between different types of ACEs and perceived coping ability (**A**) and psychiatric resilience (**B**) (β and 95% CI). Models were corrected for age, childhood deprivation, education level, civil status, employment status and income. *Coefficients are standardized.

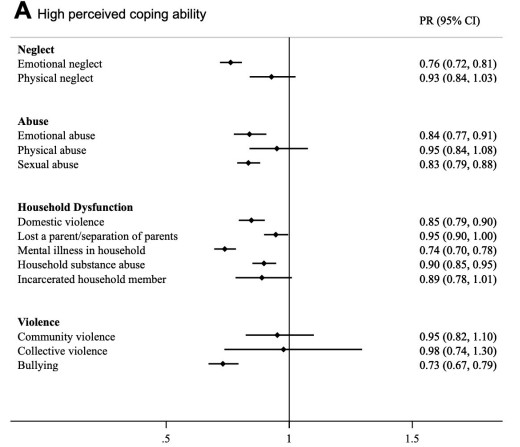
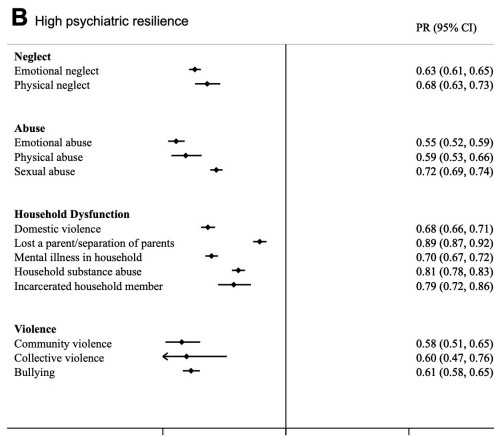

**Appendix 1—figure 6.** Prevalence Ratios (with 95% CI) of high perceived coping ability (**A**) and high psychiatric resilience (**B**) in relation to individual ACEs. Models were corrected for age, childhood deprivation, education level, civil status, employment status and income.

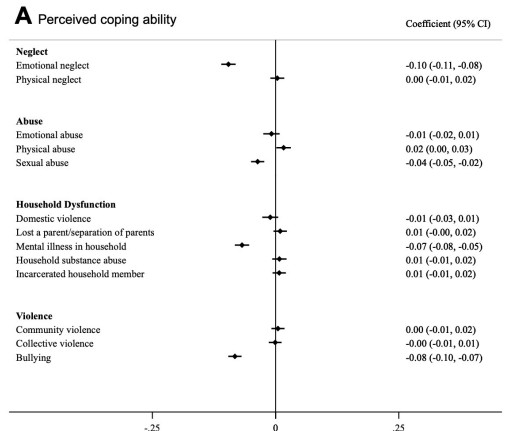
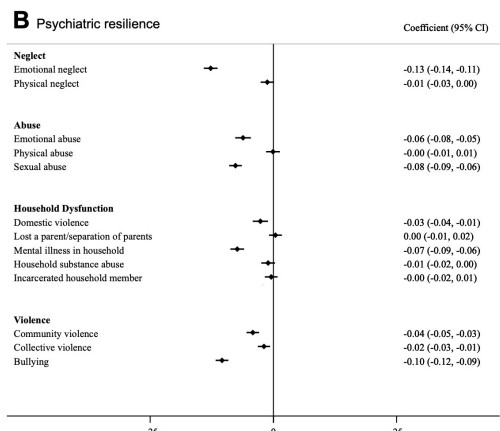

**Appendix 1—figure 7.** Associations between different types of ACEs and perceived coping ability (**A**) and psychiatric resilience (**B**) (β and 95% CI), complete case analyses (n = 19,637). Models were corrected for age, childhood deprivation, educational level, civil status, employment status, income and mutually adjusted for other ACEs. *Coefficients are standardized.

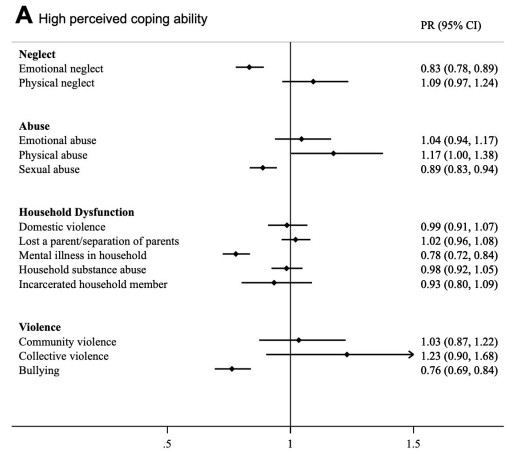
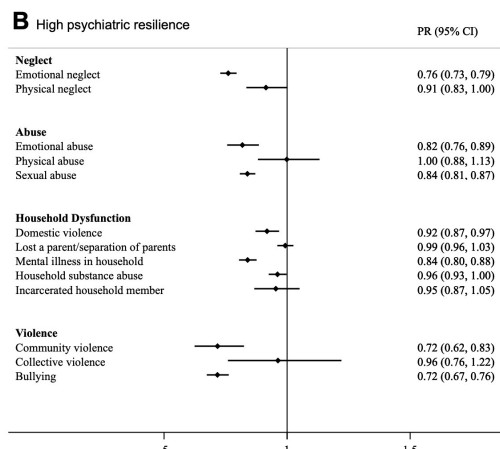

**Appendix 1—figure 8.** Prevalence Ratios (with 95% CI) of high perceived coping ability (**A**) and high psychiatric resilience (**B**) in relation to individual ACEs, complete case analyses (n = 19,637). Models were corrected for age, childhood deprivation, educational level, civil status, employment status, income and mutually adjusted for other ACEs.

