## [Editor Report]

This study aims to investigate the impact of adverse experiences during childhood on adult psychological and psychiatric resilience. Leveraging the excellent data from an ongoing cohort study on Icelandic women, the authors showed that in the face of accumulated adverse childhood events the prevalence of resilience declines, which supports earlier studies suggesting that resilience is not invincibility. The study makes an important contribution to raising awareness of the adverse childhood experiences and their impact on resiliency.

---

## [Decision Letter]

**Decision letter after peer review:**

Thank you for submitting your article "Adverse childhood experiences and resilience among adult women: a population-based study" for consideration by *eLife*. Your article has been reviewed by 3 peer reviewers, and the evaluation has been overseen by a Reviewing Editor and a Senior Editor. The following individual involved in review of your submission has agreed to reveal their identity: Toni Fleischer (Reviewer #2).

As is customary in *eLife*, the reviewers have discussed their critiques with one another. What follows below is the Reviewing Editor's edited compilation of the essential and ancillary points provided by reviewers in their critiques and in their interaction post-review. Please submit a revised version that addresses these concerns directly. Although we expect that you will address these comments in your response letter, we also need to see the corresponding revision clearly marked in the text of the manuscript. Some of the reviewers' comments may seem to be simple queries or challenges that do not prompt revisions to the text. Please keep in mind, however, that readers may have the same perspective as the reviewers. Therefore, it is essential that you attempt to amend or expand the text to clarify the narrative accordingly.

Essential revisions:

1. The conceptualisation of resilience requires a strong foundation. Although Connor-Davidson scale is well known and can be introduced as such, in this paper a new outcome-based measure was introduced, which requires laying a strong foundation on the concept of resilience. Not engaging with literature on resilience from the likes of Garmezy, Rutter, Masten, Werner, and others might have contributed to the conceptual weakness. An example might be that the authors refer to Connor- Davidson measure as perceived coping ability while the original authors meant their scale as a measure of psychological resilience in the face of perceived stress.

Although operationalisation of resilience varies, there is a common understanding that resilience is flourishing despite adversity. In this conceptualisation, establishing the role of an exposure as adversity is of paramount importance. In this paper, the psychiatric resilience is operationalised as reduction of psychiatric morbidity in the presence of trauma. However, the role of childhood events and adversities in developing psychiatric morbidity cannot be established in the current data where everyone has faced at least one trauma, thus leading to the absence of a counterfactual.

2. A second issue of concern raised is the lack of detail provided on childhood environment, beyond ACEs. While we appreciate that this information may not exist, this should be explained in the limitations of the work. Moreover, the theoretical basis for how the experience of ACEs may affect resilience was not adequately considered. Do ACEs directly affect resilience, or do these issues share risk factors in common? There are a couple of major possibilities in which ACEs could affect resilience. Firstly, those with not or low experience of ACEs may have simply been born into more stable and nurturing home environments or socioeconomically better-off households, and their relatively more favourable resilience might be reflective of this. Secondly, those with high resilience might have suppressed recall of ACEs and simply have not dwelled on them or recall them as much – 'letting go' or 'forgetting' is a resilience strategy described by some. Thirdly, ACEs might directly affect resilience. We suggest that the authors revisit their questions and present a diagram of their proposed causal model, with possible alternative pathways drawn.

3. The author should provide more information, comparing people who participated and people who did not participate in the study. Over 100,000 people were invited to participate, a third of whom chose to participate. Were there any systematic differences between those who participated and those who did not? Are these findings comparable to results from wider population, say for example against census data? Though the sample is population-based, it is unclear whether the eventual participants were representative of the source population. Are the findings generalizable? Similarly, the authors should elaborate on how missing data might have impacted the results of the study. A full third of those who participated were excluded due to missing data on one or more points. Could any sensitivity analyses be conducted to investigate this (for example, a second version of Table 1)? This is important in terms of generalizability of study findings.

4. What is the chance that those who did not answer the ACE questions had simply not had an ACE so did not answer those questions? How was this questionnaire structured?

5. Some causal language is used in the manuscript, which is not appropriate when reporting cross-sectional analyses such as these. Associations are described using the words 'increased' and 'decreased' to describe directionality in associations; 'higher' and 'lower' would be more appropriate terms.

---

## [Author Response]

Essential revisions:1. The conceptualisation of resilience requires a strong foundation. Although Connor-Davidson scale is well known and can be introduced as such, in this paper a new outcome-based measure was introduced, which requires laying a strong foundation on the concept of resilience. Not engaging with literature on resilience from the likes of Garmezy, Rutter, Masten, Werner, and others might have contributed to the conceptual weakness. An example might be that the authors refer to Connor- Davidson measure as perceived coping ability while the original authors meant their scale as a measure of psychological resilience in the face of perceived stress.

We appreciate the reviewer highlighting the need for expanding on the definition of resilience, as we acknowledge that different measures of resilience used in the literature may easily cause confusion. First, we would like to explain that we refer to resilience measured with the CD-RISC as perceived-coping ability, to distinguish this type of resilience from psychiatric resilience. Indeed, the items comprising the CD-RISC scale assess individuals’ perceptions of their abilities to adapt to change, deal with whatever comes, cope with illness and injury, handle unpleasant feelings, maintain positivity in the face of stress, and cope with obstacles. Thus, we believe that our labelling of the construct measured by CD-RISC is in line with what the authors intended their scale to measure. To clarify this, we have now adjusted the definition of perceived coping ability in the Introduction section:

“…perceived coping ability reflecting individuals’ perceptions of their ability to cope effectively with stress and adversity “(page 4, lines 104-105).

We have also adjusted this accordingly in the Methods section:

“The scale, which measures individuals’ perceptions of their ability to cope effectively with stress and adversity, such as the ability to adapt to change, achieving goals despite obstacles, and maintaining positivity in the face of stress, has demonstrated good reliability and validity” (page 7, lines 164-166).

We thank the reviewer for pointing out the works of Garmezy, Rutter, Masten and Werner. We have now extended the introduction section and cited some of their work, and also tried to emphasize that the various operationalizations of resilience remain a challenge in the field:

“And although the scientific investigation of resilience can be traced back to the 1970s (13–18), the complexity of the concept has to date contributed to varying definitions and measurement approaches (19).” (page 4, lines 100-102).

The work of Garmezy, Rutter, Masten and Werner (among others) has shaped the field of resilience, for example by setting the focus on adversity exposure as a pre-requisite for resilience. Therefore, we believe that our definitions of resilience are indeed influenced by their work.

The method to derive outcome-based psychiatric resilience, used in our study, has previously been used in other studies (Nishimi et al., 2021; Sheerin, Stratton, Amstadter, Education Clinical Center Mirecc Workgroup, and McDonald, 2018; Stein et al., 2019), but the exact psychiatric measurements behind the phenotype vary from study to study. We have now attempted to make clearer our conceptualisation of psychiatric resilience in the introduction section by highlighting that psychiatric resilience is not measured directly but rather infered based on evidence of (i) trauma exposure and (ii) positive adaptation:

“Indeed, it is important to note that psychiatric resilience is never directly measured as it consists of two separate components, trauma exposure and positive adaptation, and is therefore indirectly ascertained based on evidence of the two components (24).” (page 4, lines 107-111).

In addition, we realise that resilience has been defined and measured in multiple ways across disciplines, and we have therefore adjusted the wording slightly where we describe our operationalizations of resilience in the Introduction section, and also clarified that the definitions and measurements we use in our study align with approaches in recent studies in adult samples (i.e. to distinguish from resilience definitions in developmental sciences). This section now reads:

“Two common contemporary approaches to operationalize the concept among adults, define resilience as…” (page 4, lines 102-103).

Although operationalisation of resilience varies, there is a common understanding that resilience is flourishing despite adversity. In this conceptualisation, establishing the role of an exposure as adversity is of paramount importance. In this paper, the psychiatric resilience is operationalised as reduction of psychiatric morbidity in the presence of trauma. However, the role of childhood events and adversities in developing psychiatric morbidity cannot be established in the current data where everyone has faced at least one trauma, thus leading to the absence of a counterfactual.

We agree with the reviewer and many others (Kalisch et al., 2017; Luthar, Cicchetti, and Becker, 2000; Rutter, 2006) that resilience is generally defined as flourishing despite adversity. In fact, this is why we chose to confine our study population to women that reported at least one traumatic event during their lifetime (see description of PCL^-^5 in the manuscript). Indeed, 20% of the women did not report any adverse event during childhood, so a counterfactual condition (no ACE) is here used as a reference category in our data. However, inspired by the reviewer‘s comments we have now conducted an additional analysis where we included women with no reported lifetime trauma (see Author response table 1 yielding roughly identical results). We have not incorporated these results in the revised manuscript, since we prefer to remain consistent with our original conseptualization of resilience. We are of course willing to reconsider our position on the editors‘ request.

**Author response table 1. sa2table1:** Associations between the number of ACEs and perceived coping ability (CD-RISC) and psychiatric resilience (β and 95% CI)* Analytic sample is not restricted to women exposed to lifetime trauma.

		Perceived coping ability	Psychiatric resilience	
	N (%)	Model 1^a^	Model 2^b^	Model 1^a^	Model 2^b^
Number of ACEs					
0 ACE	4650 (23.09)	0 (ref.)	0 (ref.)	0 (ref.)	0 (ref.)
1 ACE	4628 (22.98)	-0.07 (-0.09, -0.05)	-0.05 (-0.07, -0.04)	-0.08 (-0.10, -0.07)	-0.07 (-0.09, -0.06)
2 ACE	3486 (17.31)	-0.10 (-0.11, -0.08)	-0.08 (-0.10, -0.07)	-0.13 (-0.15, -0.12)	-0.12 (-0.13, -0.10)
3-4 ACE	4021 (19.97)	-0.15 (-0.16, -0.13)	-0.12 (-0.14, -0.11)	-0.20 (-0.22, -0.19)	-0.18 (-0.20, -0.17)
≥ 5 ACEs	3350 (16.64)	-0.20 (-0.22, -0.18)	-0.15 (-0.17, -0.14)	-0.33 (-0.35, -0.32)	-0.30 (-0.31, -0.28)

*Coefficients are standardized; ^a^adjusted for age and childhood deprivation; ^b^additionally adjusted for education level, civil status, employment status and income

2. A second issue of concern raised is the lack of detail provided on childhood environment, beyond ACEs. While we appreciate that this information may not exist, this should be explained in the limitations of the work.

We agree with the reviewer that this information was lacking and we have therefore moved the description of childhood deprivation and other covariates from the Appendix to the methods section in the main text (pages 8-9, lines 208-222, 225):

"Childhood deprivation was assessed with the question: Was your family’s economic situation ever so bad that you suffered any deprivation as a consequence? For example, this could apply to deprivation of nutritious food and/or deprivation of warm clothes and appropriate footwear during the winter months, with response options ranging from 0 (never) to 4 (often).“

Indeed, (socio-economic) deprivation during childhood is an important confounder of the association between ACEs and resilience and we therefore include this factor in all models. We have now attempted to make this more clear in the Discussion section:

"The wealth of relevant data also allowed us to adjust for childhood deprivation, an important confounder when examining ACEs and adult outcomes (47,52,53), which has not been taken into account in previous ACE-resilience studies.” (page 20, lines 389-391).

Moreover, the theoretical basis for how the experience of ACEs may affect resilience was not adequately considered. Do ACEs directly affect resilience, or do these issues share risk factors in common? There are a couple of major possibilities in which ACEs could affect resilience. Firstly, those with not or low experience of ACEs may have simply been born into more stable and nurturing home environments or socioeconomically better-off households, and their relatively more favourable resilience might be reflective of this. Secondly, those with high resilience might have suppressed recall of ACEs and simply have not dwelled on them or recall them as much – 'letting go' or 'forgetting' is a resilience strategy described by some. Thirdly, ACEs might directly affect resilience. We suggest that the authors revisit their questions and present a diagram of their proposed causal model, with possible alternative pathways drawn.

We thank the reviewer for raising this important issue. We agree that ACEs can be associated with resilience through different pathways. We have now added a proposed causal model in the Appendix (Appendix Figure 3) and addressed the points raised by the reviewer in the discussion. In the diagram we mainly depict three possible pathways (most of which were kindly suggested by the reviewer): (i) ACEs could directly influence resilience (the straight line in the diagram), (ii) the association could be confounded by age and childhood environment, e.g. socio-economic deprivation in childhood, and (iii) the association could be mediated by various factors, e.g. social support.

From the methods section (page 8, lines 204-208):

"We summarised variables with a conceptual rationale for being associated both with ACEs and perceived coping ability or psychiatric resilience in Appendix Figure 3. We considered as covariates, age (at responding) and childhood deprivation (potential confounders), as well as educational level, employment status, civil status and current monthly income at responding (potential mediators).”

With respect to confounding, we certainly agree with the reviewer that those with 0 or few ACEs may have grown up in socioeconomically better-off households which in turn could have influenced their resilience in adulthood. As stated above, we therefore adjusted for childhood deprivation in all analyses which should to some extent alleviate such a concern. However, residual confounding of childhood conditions may contribute to our estimates and we have therefore commented on that in the limitation section in the discussion (page 20, lines 397-399).

"Although we adjusted for an array of important confounding factors (including childhood deprivation), we cannot exclude the possibility that unmeasured or residual confounding may contribute to our results.“

We also acknowledge the reviewers second point that resilient individuals may be less likely to recall childhood adversity which could account, at least partly, for our obtained results. Although previous research indicates that retrospective assessments of ACEs have acceptable validity, we have no possibility to address whether if and how such systematic recall bias influences our results. We therefore comment on this source of bias in our limitations section (page 20-21, lines 399-402):

“In addition, ACEs were retrospectively reported and are thus subject to potential recall bias. However, previous studies have shown acceptable validity for retrospective assessments of ACEs (52–55) although they may be influenced by current mental health status or response style (52). Yet, the similar results obtained from our sensitivity analyses excluding individuals with extreme values on the happiness assessment, reduce concerns that our results are due to a particular response style.”

Finally, we extended the discussion on how ACEs may potentially influence adult resilience (page 19-20, lines 374-379):

"Indeed, one possible mechanism through which ACEs could influence resilience, are functional outcomes in adulthood (e.g. social, financial and/or educational functioning). Previous research has found that ACEs are associated with greater risk of poor educational and financial outcomes, as well as poor social functioning (46,47), both adult factors that have previously been associated with resilience (22,29,48). However, in the current study, effect sizes only diminished slightly when we additionally adjusted for adult socio-demographic factors.”

3. The author should provide more information, comparing people who participated and people who did not participate in the study. Over 100,000 people were invited to participate, a third of whom chose to participate. Were there any systematic differences between those who participated and those who did not? Are these findings comparable to results from wider population, say for example against census data? Though the sample is population-based, it is unclear whether the eventual participants were representative of the source population. Are the findings generalizable?

We agree with the reviewer and have now added a figure to the Appendix where we contrast the distribution of age, educational level, monthly wages, and geographical region of participants of the SAGA cohort with the total female population of Iceland according to available data from Statistics Iceland (Appendix Figure 1). We previously referred to our website where this information is presented in Icelandic (www.afallasaga.is/nidurstodur), and now cite the website in the figure legend. We also added a sentence to the Methods section: (page 5, lines 132-134).

“The participants of the SAGA cohort represent the general Icelandic female population in terms of age, education level, geographical location and monthly wages (Appendix Figure 1).”

Similarly, the authors should elaborate on how missing data might have impacted the results of the study. A full third of those who participated were excluded due to missing data on one or more points. Could any sensitivity analyses be conducted to investigate this (for example, a second version of Table 1)? This is important in terms of generalizability of study findings.

We thank the reviewer for this important point. We completely agree that it is unfortunate to miss a full third of participants by stringently defining the analytic sample based on complete cases only. Thus, we have now performed multiple imputation to handle missing data on the five scales used to derive psychiatric resilience (PHQ-9, GAD-7, PCL^-^5, PSQI-A and binge drinking) and on the ACE-IQ scale. We imputed data for participants with less than 25% missing answers on each scale, which increased the sample size of the analytic sample from 19,637 (complete dataset) to 26,198 (imputed dataset). We have now added a description of the imputations to the Methods section (page 9-10, lines 227-237), and also updated the Results section with the imputed dataset: Table 1, Table 2, Table 3, Table 4, Figure 1 and Figure 2, and in text (page 16, lines 307-308, 315-319). To check whether the imputation altered our findings, we repeated the main analysis in the original dataset with complete data (Appendix Tables 7 and 8 and Appendix Figures 7 and 8). Overall, the results based on the complete dataset and the results based on the imputed dataset were similar in terms of both effect sizes and significance levels (described on page 18, lines 398-401).

From the methods section (page 9-10, lines 227-237):

“Multiple imputation

The ACE-IQ scale and the five psychopathology scales (PHQ-9, GAD-7, PCL-5, PSQI-A and binge drinking) used to derive psychiatric resilience had missing values which resulted in a reduced sample size (see Appendix Figure 4). We used multiple imputation (MI) to replace missing data with m=20 rounds of imputations, using predictive mean matching (45). We imputed data for participants who responded to more than 75% of items on each scale and then calculated the total score for the scales. The subsequent analyses (described below) were conducted using the imputed dataset. For comparison, the main analyses were repeated in the original dataset with complete data. ”

From the Results section (page 18, lines 337-340)

“Overall, the results of the complete case analyses were similar to the results using multiple imputation (main analyses) in terms of both effect sizes and confidence intervals. See Appendix Tables 7 and 8 (number of ACEs), and Appendix Figures 7 and 8 (ACE subtypes) for the complete case analyses.”

4. What is the chance that those who did not answer the ACE questions had simply not had an ACE so did not answer those questions? How was this questionnaire structured?

Thank you for this comment. We do not believe there is a big chance of this, since participants had the opportunity to indicate they had never experienced an ACE on each item (i.e. by either answering "Never“ or "No“). However, we agree that our description of the ACE-IQ questionnaire and its scoring was not ideal, and we have now added a sentence to the methods section describing the scoring of the ACE-IQ questionnaire more thoroughly (page 6, lines 151-154).

"Response options varied between items and items were either answered on a 5-point scale ranging from 0 (never) to 4 (always), on a 4-point scale ranging from 0 (never) to 3 (many times) or answered dichotomously 0 (no) and 1 (yes).“

5. Some causal language is used in the manuscript, which is not appropriate when reporting cross-sectional analyses such as these. Associations are described using the words 'increased' and 'decreased' to describe directionality in associations; 'higher' and 'lower' would be more appropriate terms.

We agree with the reviewer and have now changed the wording accordingly throughout the manuscript.

References

Kalisch, R., Baker, D. G., Basten, U., Boks, M. P., Bonanno, G. A., Brummelman, E.,... Kleim, B. (2017). The resilience framework as a strategy to combat stress-related disorders. *Nature Human Behaviour, 1*(11), 784-790. doi:10.1038/s41562-017-0200-8

Luthar, S. S., Cicchetti, D., and Becker, B. (2000). The construct of resilience: a critical evaluation and guidelines for future work. *Child development, 71*(3), 543-562. doi:10.1111/1467-8624.00164

Nishimi, K., Choi, K. W., Cerutti, J., Powers, A., Bradley, B., and Dunn, E. C. (2021). Measures of adult psychological resilience following early-life adversity: how congruent are different measures? *Psychol Med, 51*(15), 2637-2646. doi:10.1017/s0033291720001191

Rutter, M. (2006). Implications of resilience concepts for scientific understanding. *Ann N Y Acad Sci, 1094*, 1-12. doi:10.1196/annals.1376.002

Sheerin, C. M., Stratton, K. J., Amstadter, A. B., Education Clinical Center Mirecc Workgroup, T. V. M.-A. M. I. R., and McDonald, S. D. (2018). Exploring resilience models in a sample of combat-exposed military service members and veterans: a comparison and commentary. *European journal of psychotraumatology, 9*(1), 1486121-1486121. doi:10.1080/20008198.2018.1486121

Stein, M. B., Choi, K. W., Jain, S., Campbell-Sills, L., Chen, C. Y., Gelernter, J.,... Ursano, R. J. (2019). Genome-wide analyses of psychological resilience in U.S. Army soldiers. *Am J Med Genet B Neuropsychiatr Genet, 180*(5), 310-319. doi:10.1002/ajmg.b.32730